# Tele-Exercise in Non-Hospitalized versus Hospitalized Post-COVID-19 Patients

**DOI:** 10.3390/sports10110179

**Published:** 2022-11-16

**Authors:** Vasileios T. Stavrou, Kyriaki Astara, Pavlos Ioannidis, George D. Vavougios, Zoe Daniil, Konstantinos I. Gourgoulianis

**Affiliations:** 1Laboratory of Cardio-Pulmonary Testing and Pulmonary Rehabilitation, Respiratory Medicine Department, Faculty of Medicine, University of Thessaly, 41110 Larissa, Greece; 2Department of Neurology, 417 Army Equity Fund Hospital (NIMTS), 115 21 Athens, Greece; 3Department of Neurology, Faculty of Medicine, University of Cyprus, Lefkosia 2408, Cyprus

**Keywords:** tele-exercise, dyspnea, fatigue, fitness, well-being

## Abstract

The purpose of our study was to investigate the effect of tele-exercise (TE) performed for 4 consecutive weeks on fitness indicators in hospitalized post-COVID-19 patients versus non-hospitalized patients. Forty COVID-19 survivors were included, and divided into two groups: non-hospitalized versus hospitalized. Body composition, anthropometric characteristics, pulmonary function tests, single-breath diffusing capacity for carbon monoxide, 6-min walk tests (6MWT) and handgrip strength tests were recorded before and after a TE regimen (3 sessions per week, 60 min each session, warm-up and cool-down with mobility exercises, aerobic exercise such as walking outdoors, and multi-joint strength exercises). Following TE, the 6-min walk distance and handgrip were increased in both groups, with a greater observed response in the non-hospitalized group (6MWT: 32.9 ± 46.6% vs. 18.5 ± 14.3%, *p* < 0.001; handgrip: 15.9 ± 12.3% vs. 8.9 ± 7.6%, *p* < 0.001). Self-assessed dyspnea and leg fatigue were reduced in both groups, while a higher percentage of reduction was observed in the non-hospitalized group (dyspnea: 62.9 ± 42.5% vs. 37.5 ± 49.0%, *p* < 0.05; leg fatigue: 50.4 ± 42.2% vs. 31.7 ± 45.1%, *p* < 0.05). Post- vs. pre-TE arterial blood pressure decreased significantly in both groups, with the hospitalized group exhibiting more prominent reduction (*p* < 0.001). Both groups benefited from the TE program, and regardless of the severity of the disease the non-hospitalized group exhibited a potentially diminished adaptative response to exercise, compared to the hospitalized group.

## 1. Introduction

COVID-19 represents a multisystemic disease that may adversely affect the survivor’s health in multiple domains well beyond its acute phase. In COVID-19 survivors, a multitude of symptoms and conditions may arise that may be ameliorable to rehabilitation, including persistent respiratory symptoms, dyspnoea, fatigue and limited functional capacity [1]. Aside from its associations with obesity, alterations in body composition and specifically sarcopenia may also complicate the overall phenotype with or without association with other comorbidities [2,3], in a similarly bivalent manner.

The benefits of pulmonary rehabilitation are well documented, representing an established recovery approach that should be available to all patients [4]. However, pulmonary rehabilitation remains grossly underutilized by suitable patients worldwide [5]. Tele-rehabilitation involves the use of information and communication technologies to provide rehabilitation services remotely to people in their homes, while unsupervised pulmonary rehabilitation represents a telemedicine approach that has gained impetus during the COVID-19 pandemic [6]. Previous studies have reported that the tele-rehabilitation program versus standard rehabilitation shows similar results in patients with chronic respiratory disease, and is equally as effective as hospital-based rehabilitation [7].

Tele-exercise is used to maintain the physical fitness of community-dwelling people during isolation and when outpatient rehabilitation services are not available, such as during the COVID-19 pandemic period [8], while unsupervised pulmonary rehabilitation programs in previously hospitalized COVID-19 patients are an effective and beneficial practice for promoting exercise and symptom recovery following post-COVID-19, as well as a novel approach concerning the treatment of persistent fatigue induced by SARS-CoV-2 infection [9].

Exercise represents the core of pulmonary rehabilitation programs, comprising 76–100% of programs internationally [10], aiming towards a comprehensive functional recovery for respiratory disease patients [11]. Exercise reduces symptoms, increases functional ability, and improves quality of life, while tele-exercise in particular has been utilized in addressing the needs of COVID-19 survivors [9]. The average duration of international pulmonary rehabilitation programs ranges from 6 to 9 weeks, with some providing ongoing maintenance programs.

There is insufficient information in the literature about the impact of shorter duration TE and its impact on fitness indicators, strength, stamina and body composition [5,9,12]. Therefore, the purpose of our study was to investigate the effect of tele-exercise (TE) performed for 4 consecutive weeks on fitness indicators in hospitalized post-COVID-19 patients versus non-hospitalized patients.

## 2. Materials and Methods

### 2.1. Participants

Forty COVID-19 survivors volunteered for this study. They were divided into two groups: non-hospitalized (i.e., mild or moderate COVID-19) versus hospitalized, according to national guidelines (Table 1). Subjects were recruited (Figure 1) consecutively between September 2021 and November 2021 (Delta variant). For the sample size calculation of this study, a power of 86% and a confidence interval of 95% were adopted, with an estimated value for a type 1 error of 5% as there was no previous study investigating the effect of tele-exercise between non-hospitalized and hospitalized patients with COVID-19. As a final result, a value of 14 patients was obtained. However, because this is a new method of exercise, we recruited more patients (Figure 1).

Inclusion criteria were: age ≥20 to ≤60 years, without absolute and relative contraindications for a 6-min walk distance test [13], BMI ≤ 35 kg/m^2^, daily physical strain due to working ≤ 3 h/day [14], and weekly exercise ≤ 100 min per week [15], comorbidity free, without any form of musculoskeletal disability which could impair maximum exercise capacity [9,16], without active self-reported symptoms (chest pain, fatigue and/or dyspnea) [9], and without laboratory confirmed, incident respiratory disease (forced expiratory volume in 1st sec ≥85% of predicted and single-breath diffusing capacity for carbon monoxide >75% of predicted). Moreover, the hospitalized patient group was selected with the additional criterion of a two-month interval since discharge from hospital. Additional inclusion criteria for this group were: no longer require O_2_, without fever for consecutive 48-h period, stable patients, without supplemental O_2_ [9,17].

The study’s protocol was approved by the Institutional Review Board/Ethics Committee of the University Hospital of Larissa, Greece (approval reference number: No. 3952/03-11-2021). All patients provided written informed consent, in accordance with the Helsinki declaration, and personal data according to the European Parliament and the Council of the European Union.

### 2.2. Data Collection

Body composition and anthropometric characteristics were recorded according to Stavrou et al. [9] using Tanita MC-980 (Arlington Heights, IL, USA). Pulmonary function tests and single-breath diffusing capacity for carbon monoxide (Master Screen, VIASYS HealthCare, Hoechberg, Germany) were recorded according to the latest ATS/ERS guidelines [18]. The 6-min walk test (6MWT) was performed according to the ATS guidelines [19]. Other measurements included blood pressure (Sphygmomanometer Mac Check 501, Tokyo, Japan), arterial O_2_ saturation (SpO_2_) and heart rate (HR) (Nonin 9590 Onyx Vantage, Plymouth, MN, USA). Dyspnea and lower limb fatigue was assessed via the CR-10 Borg scale [20] before and at the end of 6MWT. SpO_2_ and HR [9,17] were recorded every minute of the test, as well as the total distance during the 6MWT. Handgrip strength was assessed by an electronic dynamometer (Camry, EH 101, South El Monte, CA, USA) performing one maximum isometric effort for 5 s with both hands alternately and in random order. All patients reported their dominant upper limb before the trials [17]. The same procedure was repeated after a 4-week tele-exercise intervention period.

### 2.3. Tele-Exercise Program

The tele-exercise program lasted 4 weeks (12 sessions), with each patient taking part in 3 training sessions per week. The duration of each training session was 60 min and included:∘Warm-up: 5 min mobility exercises (child’s pose/prayer stretch, doorway stretch, quadriceps stretch) for upper and lower limbs, 2 sets for 20 s each exercise with 20 s rest;∘Aerobic exercise: a 30 min continuous walk outdoors (flat and hard surface), and every five minutes patients checked their heart rate and oxygen saturation and subsequently recorded the total distance covered. The intensity was calculated according to HR_peak_ during 6MWT (approximately on 90 to 110% of HR_peak_) and patients self-reported feelings of dyspnea and leg fatigue according to CR-10 Borg scales (approximately on 5 to 6 score, respectively);∘Strength exercise: 20 min multi-joint strength exercises with body weight (chair lunges, side lateral raises, seated leg raises and squats), 3 sets for 8–12 repetitions with 40 s rest. The intensity was calculated according to the Borg scale (approximately on a 5 to 6 score from feelings of dyspnea and on a 4 to 5 score from feeling leg fatigue);∘Cool-down: 5 min mobility exercises (child’s pose/prayer stretch, doorway stretch, quadriceps stretch) for upper and lower limbs, 2 sets for 20 s each exercise with 20 s rest.

Each patient received a video with instructions for proper tele-exercise from the platform ustep (https://ustep.gr/, accessed on 10 November 2021), and adherence to the program was determined via one online video call per week. Each video call was regarding possible difficulties in performing exercises and troubleshooting. Each patient chose the time of day to exercise (between 9:00 am and 9:00 pm), while the exercise days were Monday, Wednesday and Friday. Every Friday patients exercised online with tele-supervision, and the other two days were unsupervised. The evaluated parameters (HR, SpO_2_, dyspnea and leg fatigue) were uploaded onto the ustep platform at the end of each session.

### 2.4. Statistical Analysis

A two-way repeated measures ANOVA (group x time) was used to determine statistically significant interaction effects in dependent variables before and after tele-exercise periods and between groups. A Bonferroni post-hoc test was used to locate any differences between means. The IBM SPSS 21 (SPSS Inc., Chicago, IL, USA) software was used for all analyses. The level of significance was set at *p* < 0.05. Data are presented as mean ± standard deviation and percentage (%) where appropriate.

## 3. Results

The post tele-exercise performance of the 6-min walk distance (6MWD) was improved in both groups compared with the baseline values (6MWD: F1, 38 = 36.8, *p* < 0.001, Figure 2). An analysis of covariance was used to control for baseline performance inequality in 6MWD between the groups. A greater percentage and distance improvement were observed in the hospitalized group compared to non-hospitalized in the 6MWD (32.9 ± 46.6% vs. 18.5 ± 14.3%, Figure 2).

At baseline, SpO_2_ at the end of the 6MWT appeared significantly reduced compared to the values at the beginning of the test in both groups, while a higher percentage reduction in SpO_2_ was observed in the non-hospitalized group (−2.9 ± 1.0% vs. −2.3 ± 2.0%, F1, 38 = 54.1, *p* < 0.001, Figure 3). After the tele-exercise period a significant drop in SpO_2_ appeared compared to the values at the start of the test in both groups, while a higher percentage of reduction was observed in the non-hospitalized group (−2.3 ± 1.7% vs. −1.8 ± 2.0%, F1, 38 = 53.2, *p* < 0.001, Figure 3).

At baseline, Borg scales at the end of the 6MWT appeared significantly increased compared to the values at the start of the test in both groups, while higher percentages of dyspnea and leg fatigue parameters were observed in the non-hospitalized group (dyspnea: 77.7 ± 24.1% vs. 33.3 ± 47.1%, F1, 38 = 12.2, *p* < 0.05, Figure 4; leg fatigue: 77.1 ± 17.7% vs. 10.0 ± 31.6%, F1,38 = 13.2, *p* < 0.001, Figure 5). After the tele-exercise period a significant reduction in leg fatigue was experienced in both groups, while a higher percentage of reduction in dyspnea and leg fatigue was observed in the non-hospitalized compared to the hospitalized group (dyspnea: 62.9 ± 42.5% vs. 37.5 ± 49.0%, F1, 38 = 12.3, *p* < 0.05, Figure 4; leg fatigue: 50.4 ± 42.2% vs. 31.7 ± 45.1%, F1, 38 = 11.9, *p* < 0.05, Figure 5).

At baseline, arterial pressure (SBP, DBP and MAP) at the end of the 6MWT appeared significantly increased compared to the values at the start of the test in both groups, while a higher percentage of increase was observed in the hospitalized compared to non-hospitalized group (F1, 38 = 65.2, *p* < 0.001, Table 2). Following the tele-exercise period, a significant reduction in post- vs. pre-TE values was observed in the groups, with the hospitalized group exhibiting a more prominent reduction (F1, 38 = 42.1, *p* < 0.001, Table 2). Heart rate differences were not noted at baseline or following TE. Furthermore, no interaction was observed between groups and measurements (F1, 38 = 0.204, *p* > 0.05, Table 2).

Post-TE handgrip strength was improved in both groups compared with the baseline values (F1, 38 = 36.8, *p* < 0.001, Figure 6). Analysis of covariance was used to control for baseline performance inequality in handgrip strength between the groups. A greater percentage improvement was observed in the non-hospitalized compared to hospitalized group in handgrip strength (15.9 ± 12.3% vs. 8.9 ± 7.6%, Figure 6). There was no difference between the baseline and post-tele-exercise period, nor between group values in anthropometric and body composition variables (*p* > 0.05).

## 4. Discussion

The results have shown that in both groups, exercise was beneficial for post-COVID rehabilitation. In fact, the combination of an aerobic-anaerobic training program, with its distinct effects, provides a holistic approach to total physical rehabilitation. What matters, though, is the influence of the severity of COVID-19 in the post-COVID interval.

Despite the overall better performance after the designated training program, the response to exercise was different between the groups. The hospitalized group exhibited a more prominent drop in SpO_2_ and an increase in BP at the end of the 6MWT. This could signify a more optimal autonomic adaptation to exercise stimuli. Post-COVID patients suffering from impaired oxygenation and muscle trophicity were hindered, especially if they had been previously hospitalized [17]. As a compensatory, blood pressure increased linearly with its constituent of cardiac output, in order to maintain adequate perfusion in muscles during exercise [21]. Therefore, the hospitalized group may display wider margins of improvement during rehabilitation.

Previous study by our research team reported that the pathophysiology underlying lasting hypoxia secondary to COVID-19 may be ventilation–perfusion mismatch [9]. Exercise widens the alveolar–arterial PO_2_ difference, due to VA/Q, and interstitial pulmonary edema, inadequate ventilatory response, and/or alveolar–capillary diffusion, resulting in the further limitation of O_2_ transport [9].

Accordingly, the non-hospitalized group seems to suffer from more prominent post-COVID symptoms, perturbing their potential to benefit from rehabilitation. The scores from the Borg scale could indirectly support this result. The non-hospitalized group claimed more persistence in perceived intensity of exercise, both aerometric and strength. This could be attributed to the mismatch between the most frequent phenotype of SARS-CoV-2 infection and the national guidelines of severity assessment. The criteria of hospital admission were based on the presence of fever among other signs, whereas in data-driven analysis for COVID-19 phenotypes, it was shown that febrility covered the smallest portion [22]. Previous study by our research team reported that prolonged periods of bed rest have been shown to induce substantial changes in body composition and are accompanied by overall metabolic decline and affect potentially all body muscles [17]. COVID-19 survivors may experience acute sarcopenia associated by lockdown, which leads to long-post-COVID-19 syndrome [17]. These associations are reflected in our results and should be addressed by targeted rehabilitation. Our results show greater percentage improvement in the non-hospitalized compared to hospitalized group in the handgrip strength test (15.9 ± 12.3% vs. 8.9 ± 7.6%).

The putative mechanism by which rehabilitation ameliorates this damage has been previously described in interstitial lung disease, where the dyspnea reduced after exercise rehabilitation programs [23]. Patients with chronic lung diseases had exercise intolerance due to reduced breathing efficiency that resulted from the deteriorating ventilatory mechanics on one hand and the increased ventilatory requirement on the other hand [24]. In our study the impact of the 30 min continuous walk was on feelings of dyspnea of 5 to 6 score (Borg scale), aiming to improve the limiting factor of dyspnea with the process of central desensitization of dyspnea.

Our study should be interpreted within the context of its strengths and limitations. An important limitation of our study is that rehabilitation was performed within a relatively limited duration. However, despite the limited duration of the rehabilitation program, both groups showed significant improvement in 6MWD and handgrip strength. Additionally, the rest of the parameters of fitness indicators, when examined, showed a trendline towards enhanced physical condition. It should be noted that the rehabilitation program was unsupervised, allowing a greater degree of freedom to both the clinician and the patient. Telemedicine has gained popularity during the COVID-19 pandemic, as access to hospitals and outpatient departments was limited [25]. Another important limitation is that per the study protocol, patient groups older than 60 and with BMI greater than 35 were not studied. While these patient groups were not the focus of our study, similar studies could explore the potential benefit of TE. A final limitation is that as the study involved survivors infected with the delta variant, an analysis of the variant as a variable could not be performed. Studies with study samples that extend to the initial infection and Omicron could thus provide more information on whether variants specifically had any impact on rehabilitation in general. Finally, our findings should be interpreted within the context of practical significance. As such, while statistically significant, differences in heart rate that do not exceed physiological thresholds (i.e., brady–tachycardia or clinically significant fluctuations) [26] are unlikely to inform clinical decision making; the same could be said about the oscillations noted in the arterial blood pressure. Differences in handgrip strength and 6-min walking distance, however [27,28], may be more reliable in informing decisions and guiding rehabilitation. Notably, our findings are corroborated by studies with larger sample sizes [29] and indicate that COVID-19 survivors benefit in all aspects of physical fitness.

## 5. Conclusions

In conclusion, a tele-pulmonary rehabilitation program had a beneficial effect in all COVID-19 survivors, with more prominent results having been observed in the hospitalized group. Our results put emphasis on the need for rehabilitation to be addressed for all COVID-19 survivors. Furthermore, it is crucial to address this need in an individualized manner. Our results support a clear difference in the needs and adaptation of different severity groups in COVID-19 survivors, and potentially imply a second order question: that of the necessary duration and follow-up for rehabilitation programs in the long-COVID setting. An important implication of our study is that as TE is shown to confer a beneficial effect, it could provide a cost-effective and potentially widely disseminated alternative to on-site rehabilitation practice.

## Figures and Tables

**Figure 1 sports-10-00179-f001:**
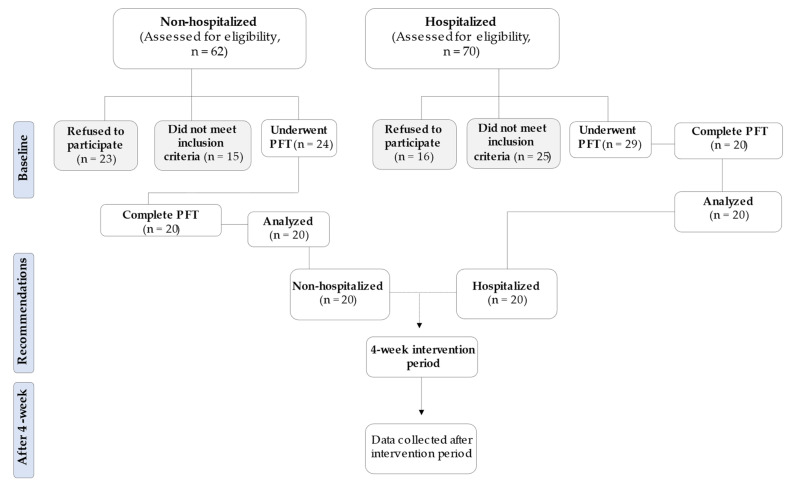
Flow study diagram. Non-hospitalized: 37.1% of eligible patients refused to participate with justifications of “not enough time”, “I cannot make it” and “I feel tired/exhausted”. A total of 24.0% of eligible patients did not meet inclusion criteria such as self-reported symptoms and incident respiratory disease. Four patients were excluded after PFT due to high blood pressure during 6MWT (systolic blood pressure > 200 mmHg). Hospitalized: 22.8% of eligible patients refused to participate with justifications of “dizziness and headache” and “discomfort feeling”. A total of 35.7% of eligible patients did not meet further inclusion criteria such as self-reported highly dyspnea symptoms, incident respiratory disease, comorbidities and musculoskeletal disabilities. Nine patients were excluded after PFT: one patient experienced an 8% desaturation after the first min of 6MWT, three patients showed high blood pressure during 6MWT (systolic blood pressure > 200 mmHg) and five patients showed, during 6MWT, staggering, high diaphoresis, intolerable dyspnea and leg cramps. Abbreviations: PFT = physical fitness test.

**Figure 2 sports-10-00179-f002:**
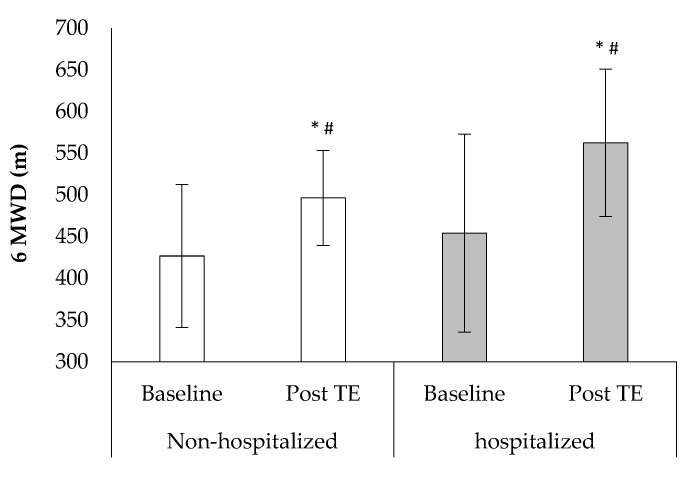
Performance changes for 6-min walk distance (6MWD) at baseline and after the tele-exercise period (post-TE) in non-hospitalized and hospitalized groups. * *p* < 0.05 between baseline and post-TE, # *p* < 0.05 between groups.

**Figure 3 sports-10-00179-f003:**
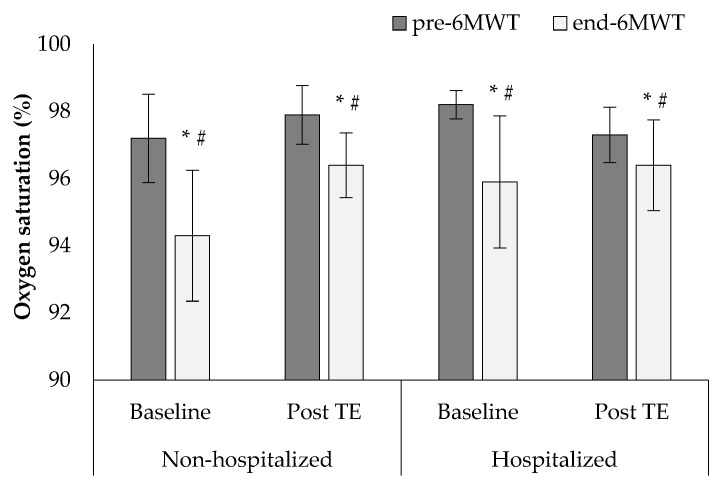
Changes in O_2_ saturation (SpO_2_) before and at the end of the 6-min walk test (6MWT) at baseline and after the tele-exercise period (post-TE) in non-hospitalized and hospitalized groups. * *p* < 0.05 between baseline and post-tele-PR, # *p* < 0.05 between groups.

**Figure 4 sports-10-00179-f004:**
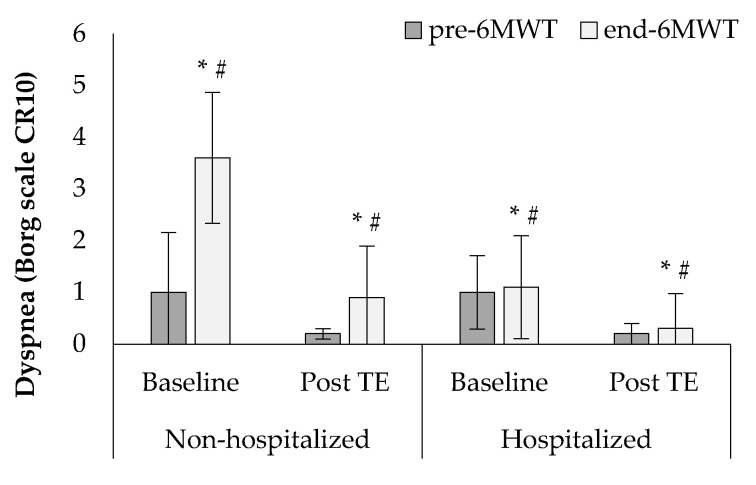
Changes in dyspnea at baseline and after the tele-exercise period (post-TE) in non-hospitalized and hospitalized groups. * *p* < 0.05 between baseline and post-TE, # *p* < 0.05 between groups.

**Figure 5 sports-10-00179-f005:**
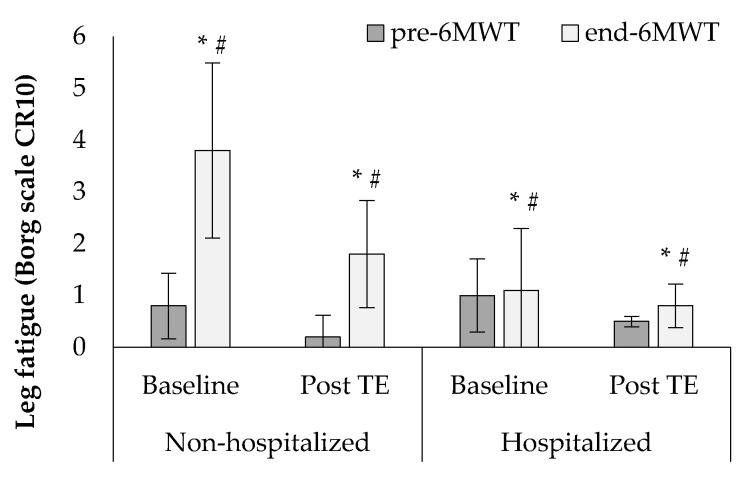
Changes in leg fatigue at baseline and after the tele-exercise period (post-TE) in non-hospitalized and hospitalized groups. * *p* < 0.05 between baseline and post-TE, # *p* < 0.05 between groups.

**Figure 6 sports-10-00179-f006:**
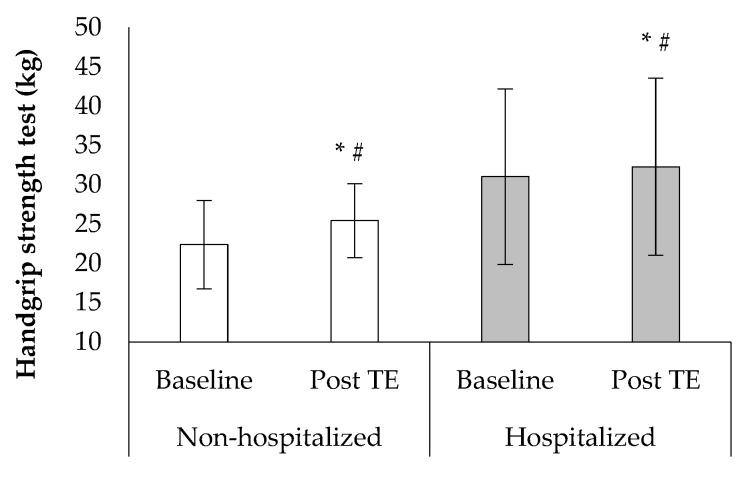
Changes in handgrip at baseline and after the tele-exercise period (post-TE) in non-hospitalized and hospitalized groups. * *p* < 0.05 between baseline and post-TE, # *p* < 0.05 between groups.

**Table 1 sports-10-00179-t001:** Body composition results between non-hospitalized and hospitalized post-COVID-19 patients. Data are expressed as percent, mean ± standard deviation.

		Non-Hospitalized	Hospitalized
Age	yrs	44.3 ± 12.2	48.9 ± 8.1
Gender (M)	*n* (%)	10 (50%)	10 (50%)
Body mass index	kg/m^2^	25.5 ± 5.3	29.0 ± 2.9
Body surface area	m^2^	1.7 ± 0.4	2.0 ± 0.2
Body fat	%	25.3 ± 7.2	31.9 ± 11.5
Total body water	%	57.5 ± 6.3	53.1 ± 1.9
Lean body mass	kg	55.3 ± 8.2	60.4 ± 3.9
Δchest	cm	3.6 ± 0.5	4.6 ± 2.8
Physical activity	min/week	45.6 ± 9.2	49.6 ± 12.4
FEV_1_	% of predicted	101.0 ± 8.0	97.9 ± 7.1
DLCO(SB)	% of predicted	81.3 ± 3.1	79.2 ± 1.4

Abbreviations: DLCO(SB) = single-breath diffusing capacity of the lung for CO; FEV_1_ = forced expiratory volume in 1st sec; Δchest: chest circumference difference between maximal inhalation and exhalation.

**Table 2 sports-10-00179-t002:** Hemodynamic parameters between groups before and at the end of the 6-min walk test (6MWT) at baseline and after the tele-exercise period (post-TE).

			Non-Hospitalized		Hospitalized	
			Baseline	Post TE	Baseline	Post TE
Systolic blood pressure	mmHg	Pre 6MWT	115.6 ± 17.3	119.0 ± 9.5 *	138.1 ± 18.8 ^#^	126.5 ± 4.7 *^#†^
		Post 6MWT	129.5 ± 18.7	124.8 ± 9.1 *	158.8 ± 14.9 ^#^	153.5 ± 9.7 *^#†^
Diastolic blood pressure	mmHg	Pre 6MWT	81.8 ± 18.8	79.8 ± 15.0 *	83.6 ± 14.3	84.0 ± 4.6
		Post 6MWT	85.5 ± 18.2	89.2 ± 11.6	89.0 ± 11.0	89.0 ± 5.2
Mean arterial pressure	mmHg	Pre 6MWT	93.1 ± 18.0	92.9 ± 12.5	101.8 ± 15.1 ^#^	98.2 ± 3.4 *^#†^
		Post 6MWT	100.2 ± 17.3	101.1 ± 10.2	112.3 ± 11.3 ^#^	110.5 ± 4.9 ^#^
Heart rate	BPM	Pre 6MWT	79.6 ± 7.2	74.5 ± 6.9 *	79.6 ± 16.8	77.5 ± 10.4 *
		Post 6MWT	135.0 ± 17.9	123.0 ± 8.7 *	118.5 ± 19.3 ^#^	115.2 ± 18.7 *

* *p* < 0.05 between baseline and post-TE values, ^#^ *p* < 0.05 between groups, ^†^ *p* < 0.05 between pre- and post-6-min walk test (6MWT).

## Data Availability

All data are available after request.

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
