# Peer review of "Tele-Exercise in Non-Hospitalized versus Hospitalized Post-COVID-19 Patients"

_sports, 2022, doi:10.3390/sports10110179_

Round 1

Reviewer 1 Report

 This is an interesting study aiming to examine the effect of tele-exercise on fitness-related parameters in post-COVID 19 patients (hospitalized patients vs non-hospitalized patients).

I believe that the findings of the study bear of high clinical significant, while tele-exercise is a recently-developed and promising form of exercise. The paper is well-written, and I have only some comments to improve the quality of the manuscript further.

Introduction:

-Line 34: Please put in brackets the abbreviation for the 'Pulmonary rehabilitation'

-Please add a short paragraph in order to include the impact of COVID-19 on patients' functional capacity, body composition and fatigue.

Methodology:

-How many absences were allowed in the tele-exercise program? Please provide more information about adherence to the tele-exercise program.

-Please change O2 to O2

-Please provide more information about the tele-exercise program. For example, what platform did the authors use to perform the exercise program? Did the session stream in real-time (online)? Supervision during some sessions? Can the authors provide data about heart rate levels during the exercise sessions?

-Statistical analysis. I assume the authors used two-way repeated ANOVA (group X time). Please correct accordingly.

Discussion

-Please add a paragraph regarding the use of tele-exercise programs during the lockdown period and for the exercise rehabilitation post-COVID-19 patients.

- The authors should elaborate further on how they explain the differences in the examined variables between the two groups

-The authors should elaborate more on the clinical importance of their findings (for each examined parameter) and compare their results with the results of the conventional exercise or/and rehabilitation of post-COVID-19 patients.

Author Response

Response to reviewer

Paper title: Tele-Exercise in non-hospitalized versus hospitalized post-COVID-19 patients

Manuscript ID: sports-1987367

We thank the reviewer for the comments that have helped us to improve the paper. All changes have been indicated by red color within the text. Below you will find a point-by-point response to your comments.

Reviewer 1

This is an interesting study aiming to examine the effect of tele-exercise on fitness-related parameters in post-COVID 19 patients (hospitalized patients vs non-hospitalized patients). I believe that the findings of the study bear of high clinical significant, while tele-exercise is a recently-developed and promising form of exercise. The paper is well-written, and I have only some comments to improve the quality of the manuscript further.

Comments 1.-Line 34: Please put in brackets the abbreviation for the 'Pulmonary rehabilitation'

Response: Thank you for your suggestion, it has been replaced

Comments 2.-Please add a short paragraph in order to include the impact of COVID-19 on patients' functional capacity, body composition and fatigue.

Response: Thank you for your suggestion, it has been added in the “Introduction” section.

Comments 3.-How many absences were allowed in the tele-exercise program? Please provide more information about adherence to the tele-exercise program.

Response: Thank you for your suggestion, it has been added in more information in the “Tele-exercise program” section

Comments 4.-Please change O2 to O2

Response: Thank you for your suggestion, it has been replaced

Comments 5.-Please provide more information about the tele-exercise program. For example, what platform did the authors use to perform the exercise program? Did the session stream in real-time (online)? Supervision during some sessions? Can the authors provide data about heart rate levels during the exercise sessions?

Response: Thank you for your suggestion, it has been added in more information in the “Tele-exercise program” section

Comments 6.-Statistical analysis. I assume the authors used two-way repeated ANOVA (group X time). Please correct accordingly.

Response: Corrected as suggested, thank you.

Comments 7.-Please add a paragraph regarding the use of tele-exercise programs during the lockdown period and for the exercise rehabilitation post-COVID-19 patients.

Response: Thank you for your suggestion, it has been added in the “Introduction” section.

Comments 8.- The authors should elaborate further on how they explain the differences in the examined variables between the two groups

Response: Thank you for your suggestion, it has been added in the “Discussion” section

Comments 9.-The authors should elaborate more on the clinical importance of their findings (for each examined parameter) and compare their results with the results of the conventional exercise or/and rehabilitation of post-COVID-19 patients.

Response: Thank you for your suggestion, it has been added in the “Discussion” section

Reviewer 2 Report

The authors have taken on board the suggestions, however, there are still shortcomings in the discussion and conclusion. I advise:

Discussion

- Should include some hypothesis of possible mechanisms described from a physiological perspective or include some mechanism of action described. Especially those referring to strength or aerobic testing (6WT).

- Should include some comparative discussion with other studies related to your objective, explaining the differences.

- What does this study contribute? Clarify.

- Any possible application of the results described?

- Include a section on strengths and limitations.

Conclusion

In the conclusion section, state the most important result of your work. Do not simply summarize the points already made in the body, but interpret your findings at a higher level of abstraction. Show whether, or to what extent, you have succeeded in answering the need stated in the introduction (or objectives).

Author Response

Response to reviewer

Paper title: Tele-Exercise in non-hospitalized versus hospitalized post-COVID-19 patients

Manuscript ID: sports-1987367

We thank the reviewer for the comments that have helped us to improve the paper. All changes have been indicated by red color within the text. Below you will find a point-by-point response to your comments.

Reviewer 2

The authors have taken on board the suggestions, however, there are still shortcomings in the discussion and conclusion. I advise:

Comments 1.- Should include some hypothesis of possible mechanisms described from a physiological perspective or include some mechanism of action described. Especially those referring to strength or aerobic testing (6WT).

Response: Thank you for your suggestion, it has been added in the “Discussion” section

Comments 2.- Should include some comparative discussion with other studies related to your objective, explaining the differences.

Response: Thank you for your suggestion, it has been added in the “Discussion” section

Comments 3.- What does this study contribute? Clarify.

Response: We have added specific contributions in the “Conclusions” section. Thank you.

Comments 4.- Any possible application of the results described?

Response: We have added specific application in the “Conclusions” section. Thank you.

Comments 5.- Include a section on strengths and limitations.

Response: We have included this section per the reviewer’s suggestion. Thank you.

Comments 6. Do not simply summarize the points already made in the body, but interpret your findings at a higher level of abstraction. Show whether, or to what extent, you have succeeded in answering the need stated in the introduction (or objectives).

Response: Thank you for your comment. We have addressed this part in the rewritten ‘Conclusions” sections, so as to interpret rather than summarize our findings along other points made by the reviewer in a streamlined way (Comments 3,4).